# Against the Odds: Hybrid Zones between Mangrove Killifish Species with Different Mating Systems

**DOI:** 10.3390/genes12101486

**Published:** 2021-09-24

**Authors:** Waldir M. Berbel-Filho, Andrey Tatarenkov, George Pacheco, Helder M. V. Espírito-Santo, Mateus G. Lira, Carlos Garcia de Leaniz, John C. Avise, Sergio M. Q. Lima, Carlos M. Rodríguez-López, Sofia Consuegra

**Affiliations:** 1Department of Biology, University of Oklahoma, Norman, OK 73019, USA; 2Department of Biosciences, College of Science, Swansea University, Swansea, Wales SA2 8PP, UK; C.Garciadeleaniz@Swansea.ac.uk; 3Department of Ecology and Evolutionary Biology, University of California, Irvine, CA 92697, USA; tatarenk@uci.edu (A.T.); javise@uci.edu (J.C.A.); 4Section for Evolutionary Genomics, The GLOBE Institute, Faculty of Health and Medical Sciences, University of Copenhagen, 1350 Copenhagen, Denmark; george.pacheco@snm.ku.dk; 5Núcleo de Ecologia Aquática e Pesca da Amazônia, Universidade Federal do Pará, Belém 66075-110, Pará, Brazil; espiritosantohm@gmail.com; 6Laboratório de Ictiologia Sistemática e Evolutiva, Departamento de Botânica e Zoologia, Universidade Federal do Rio Grande, Natal 59078-900, Rio Grande do Norte, Brazil; tewsictio@gmail.com (M.G.L.); sergio.lima@ufrn.br (S.M.Q.L.); 7Environmental Epigenetics and Genetics Group, Department of Horticulture, College of Agriculture, Food and Environment, University of Kentucky, Lexington, KY 40506, USA; carlos.rodriguezlopez@uky.edu

**Keywords:** asymmetric introgression, *Kryptolebias*, mixed mating, reproductive isolation, self-fertilization

## Abstract

Different mating systems are expected to affect the extent and direction of hybridization. Due to the different levels of sexual conflict, the weak inbreeder/strong outbreeder (WISO) hypothesis predicts that gametes from self-incompatible (SI) species should outcompete gametes from self-compatible (SC) ones. However, other factors such as timing of selfing and unilateral incompatibilities may also play a role on the direction of hybridization. In addition, differential mating opportunities provided by different mating systems are also expected to affect the direction of introgression in hybrid zones involving outcrossers and selfers. Here, we explored these hypotheses with a unique case of recent hybridization between two mangrove killifish species with different mating systems, *Kryptolebias ocellatus* (obligately outcrossing) and *K. hermaphroditus* (predominantly self-fertilizing) in two hybrid zones in southeast Brazil. Hybridization rates were relatively high (~20%), representing the first example of natural hybridization between species with different mating systems in vertebrates. All F1 individuals were sired by the selfing species. Backcrossing was small, but mostly asymmetrical with the SI parental species, suggesting pattern commonly observed in plant hybrid zones with different mating systems. Our findings shed light on how contrasting mating systems may affect the direction and extent of gene flow between sympatric species, ultimately affecting the evolution and maintenance of hybrid zones.

## 1. Introduction

Hybridization is a major source of evolutionary innovation, with important implications for phenotypic diversification, adaptation and ultimately speciation [1,2]. Prezygotic barriers, such as spatial, temporal, and behavioral differences play an important role in regulating the extent of hybridization amongst sympatric species in nature [3]. In addition, differences in mating systems (defined as the proportion of self-fertilization versus outcrossing [4]) can largely influence the extent of hybridization, as well as the direction of gene flow and introgression [5,6]. For example, in monkeyflowers (genus *Mimulus*), differences in the mating system (predominantly selfing vs obligate outcrossing) have a strong effect as a prezygotic mechanism, leading to nearly complete reproductive isolation [6,7,8].

Natural crosses between self-compatible (SC) and self-incompatible (SI) plants generally produce viable embryos only when SI pollen (male parent) fertilizes SC styles (female parent), but not vice-versa. This pattern of unilateral incompatibility (UI) is known as the ‘SI × SC rule’ [9,10]. This rule could be explained by the weak inbreeder/strong outbreeder (WISO) hypothesis [11], according to which the higher magnitude of maternal-paternal conflict would result in outcrossers’ gametes outcompeting selfer’s gametes in crosses between species with different mating systems. In fact, sperm from outcrossers and/or species with higher levels of sperm competition are more successful at fertilization [12,13,14]. The WISO hypothesis could, therefore, explain the patterns of UI often observed in the ‘SI × SC rule’ that, despite some exceptions [9,15], is commonly observed in plants [11,16,17,18,19,20]. The ‘SI × SC rule’ has also been detected in some artificial interspecific crosses of *Caenorhabditis* nematodes, with males of outcrossing species having more competitive sperm than selfing males [14]. However, the opposite pattern was also observed in *Caenorhabditis* interspecific crosses involving other species [21], where more progeny was generated in crosses involving selfer males and outcrossing hermaphrodites than the ones in the opposite direction. These mixed results call for further research into the generality of the ‘SI × SC rule’ in animals, particularly under natural conditions.

In addition to the differential levels of genome conflict proposed by the WISO hypothesis, other factors can affect the direction of hybridization between species with different mating systems. SC organisms can self-fertilize before (prior selfing), during (competing selfing) or after (delayed selfing) outcrossing [22]. These differential selfing timings are likely to affect the opportunities for outcrossing, either by conspecific or heterospecific mates [5,23]. Prior selfing is known to represent a strong reproductive barrier for hybridization compared to the other types of selfing [24]. Contrasting mating systems can also influence the direction of introgression [5,25]. Given the possibility of both selfing and outcrossing in the F1s, the predictions about later hybrid generations in hybrid zones with SI and SC species will depend on the mating system of the hybrids themselves [19]. Although UI may emerge postzigotically according to the direction of introgression [26], in hybrid zones between SI and SC species, F1 individuals are expected to backcross more often with the SI than SC parent [5,19], creating an asymmetrical (from SC into SI) pattern of introgression. This prediction relies on the fact that highly selfing taxa usually have stronger reproductive isolation due to the low availability of gametes to outcross [22,27]. Therefore, male F1s are more likely to interact with ‘fertilizable’ eggs from the SI when compared to SC parental species [5].

Unlike plants, relatively little is known about the dynamics of hybridization between animals with different mating systems [28]. Most animals, in particular vertebrates, are dioecious (i.e., different individuals are either male or female) and therefore unable to self-fertilize [29]. *Kryptolebias hermaphroditus,* the mangrove killifish, is one of two known examples of self-fertilizing hermaphroditism in vertebrates [30]. Its populations mostly consists of selfing hermaphrodites and very rare males [31,32]. *Kryptolebias hermaphroditus* coexists with the outcrossing species *K. ocellatus* in the mangrove forests of southeast Brazil (Figure 1a and Figure 2) [33]. Intermediate levels of individual heterozygosity in some populations suggest that outcrossing, likely to only happen between males and hermaphrodites in mangrove killifishes [34], sometimes occurs in *K. hermaphroditus*, but selfing is the major mode of reproduction in the species [35]. In mangrove killifishes, selfing occurs internally, with most of the eggs laid externally being already fertilized via selfing by the hermaphrodites (e.g., prior selfing) [36,37]. The frequency of prior selfing and, consequently the availability of unfertilized eggs for outcrossing, is influenced by environmental factors, such as temperature [38]. In contrast, *K. ocellatus*’ populations are composed by males and hermaphrodites in approximately equal ratio. Although selfing cannot be fully discarded in *K*. *ocellatus,* breeding experiments only generated offspring when males and hermaphrodites were kept together [39]. In addition, population genetic studies using microsatellites across most of the species’ distribution showed a general lack of deviation from Hardy-Weinberg equilibrium across loci, strongly suggesting that *K*. *ocellatus* is an exclusively outcrossing species [33,40]. The two species are genetically very divergent, approximately 10% at mtDNA [33], which is a divergence level higher than is commonly observed in ‘well-established vertebrate species pairs [41,42]. *Kryptolebias ocellatus* is likely to be the sister species of the common ancestor between the selfing mangrove killifishes *K. hermaphroditus* and *K. marmoratus* [30,43].

The sympatry between the SI and obligately outcrossing *K. ocellatus* and the SC and predominantly selfing *K. hermaphroditus* [33] make these species a unique vertebrate system to understand the evolutionary dynamics of hybrid zones in animals with dissimilar mating systems. Here we take advantage of this system to explore, for the first time in vertebrates, some of the existing hypotheses regarding hybrid zones containing SC and SI species. First, given the low availability of gametes for outcrossing due to prior selfing in highly selfing organisms (such as *K. hermaphroditus*), hybridization rates between SC and SI taxa are expected to be generally low [5]. Second, although we cannot formally test WISO hypothesis without an experimental approach, based on its predictions, we expect that most of the natural hybrids to be sired by the outcrosser species (*K. ocellatus*), rather than by the selfer species (*K. hermaphroditus*) (Figure 1b). Alternatively, given the prior selfing nature of the SC *K. hermaphroditus*, substantial differences in mating opportunities in the wild could result in the opposite crossing direction (hereafter called the ‘mating opportunity’ hypothesis). Finally, given the differences in mating systems of the parental species (and therefore availability of fertilizable eggs), we expect that backcrossing, if existent, should be biased towards the SI parental species. In this case, we expect to find more backcross individuals between *K. ocellatus* and F1s than between *K. hermaphroditus* and F1s (Figure 1c).

## 2. Materials and Methods

### 2.1. Sampling Design and Genetic Markers 

We analyzed two different genetic datasets. First, we analyzed microsatellite data for 103 *K*. *ocellatus* individuals (all from two previous datasets collected either in 2017 [33] or 2007 [40]) and 42 *K*. *hermaphroditus* individuals (32 sampled in 2017 and genotyped in this study; 10 sampled in 2007 [40]) from four localities in south and southeast Brazil (Figure 2; Table 1). The species are sympatric in two of those locations, Guaratiba and Fundão (GUA and FUN respectively in Figure 2), at the west and east limits, respectively, of the Rio de Janeiro municipality in Brazil [32,33]. The other two localities (SFR and GUA in Figure 2) represent an area only inhabited by *K. ocellatus.* Sampling was carried out under license ICMBio/SISBIO 57145-1/2017. Fish species were identified morphologically and confirmed by cytochrome oxidase subunit I (*cox1*) barcoding [33]. For the newly sampled material (*K. hermaphroditus* sampled in 2017; Table 1) we used a set of 16 microsatellites from Mackiewicz, et al. [44]. Micro-checker v. 2.2 (van Oosterhout et al. 2004) was used to check for errors or presence of null alleles. FSTAT v. 2.9.3.2 [45] was used retrieve basic statistics from microsatellite data, which are provided in Appendix A.

Although microsatellites allowed us to genotype many individuals, the few loci analyzed could result in incomplete information about the genome-wide admixture proportions. Therefore, we expanded our sampling of genomic loci by obtaining new genomic data using a methylation sensitive genotype-by sequencing (msGBS) from pectoral-fin samples of 55 hermaphrodite individuals (33 *K. ocellatus* and 22 *K*. *hermaphroditus* sampled in 2017) (Table 1). Genomic DNA was extracted using Qiagen^®^ DNeasy Blood and Tissue kit (Qiagen^®^, Hilden, Germany) following the manufacture’s protocol. msGBS libraries were prepared as described in Kitimu, et al. [46]. In brief, extracted DNA was digested using restriction enzymes EcoRI and HpaII and ligated to sequencing adapters. A single library was produced by pooling 20 ng of processed DNA from each restriction/ligation product and amplified in eight separate PCR reactions which were pooled after amplification, size-selected (range 200–350 bp) and sequenced in a single of lane of an Illumina NextSeq500 sequencer.

Paired-end reads were demultiplexed using GBSX v 1.3 [47]. We then filtered (-qtrim r; -minlength 25) and merged the reads by individual using BBmap tools [48] mapped to the *Kryptolebias marmoratus* reference genome [49] using Bowtie 2 v. 2.2.3 and generated filtered and indexed individual BAM files with SAMtools v. 1.9 [50]. Given the closer phylogenetic proximity between *K. hermaphroditus* and *K. marmoratus* (species from the reference genome) [43], a higher number of uniquely mapped reads was found between *K. hermaphroditus* and *K. marmoratus* (average 89%), than between *K. ocellatus* and *K. marmoratus* (average 80%) (Appendix A). To call genotypes across all samples, we used ANGSD v 0.9.2.9 [51]. Given the methylation sensitivity of HpaII and the differential mapping efficacy between species, we only allowed for a maximum of 5% of missing data per loci across all samples. Single- and double-tons were removed and, using SAMtools genotype likelihood model, we estimated posterior genotype probabilities assuming a uniform prior (-doPost 2). We also used the ANGSD (-SNP_pval 1 × 10^−6^) to carry out a Likelihood Ratio Test to compare between the null (minimum allele frequency = 0) and alternative (estimated minimum allele frequency) hypotheses by using a Chi-squared distribution with one degree of freedom. More details about the library preparation and data processing are provided in the Appendix A.

These analyses produced two genomic datasets for 53 samples: dataset I with 597,733 sites, with average coverage per individual between 12.0X and 346.5X (mean 145.2X) and average missing data per individual ranging from 0% to 7.2% (mean 0.50%), and dataset II with 5477 SNPs, average coverage per individual between 12.4X and 382.6X (mean 152.9X) and average missing data per individual ranging from 0% to 4.9% (mean 0.34%) (Appendix A). A strong correlation (R^2^ = 0.93, *p* < 0.001) between the size of the 3073 scaffolds of the *K. marmoratus* reference genome and the number of SNPs from each scaffold indicated that the SNPs were evenly distributed throughout the reference genome.

### 2.2. Population Genetics and Hybridization Analysis with Microsatellite and SNP Data

To investigate the structure and direction of the potential hybrid zones between *K. ocellatus* and *K. hermaphroditus*, STRUCTURE 2.3.4 [52] was used with microsatellite data with the following parameters: 10 iterations per K (K ranging 2–10), a total of 1,000,000 MCMC, 100,000 burn-in, admixture model and independent allele frequencies. Independent STRUCTURE runs were aligned and plotted using CLUMPAK [53]. To identify the uppermost hierarchical level of genetic structure, we chose the most likely K value using second-order rate of change of likelihood ∆K method [54], implemented in in Structure Harvester [55]. For the microsatellites, genotypic associations among individuals were visualized using the factorial correspondence analysis (FCA) implemented in GENETIX v. 4.04 [56]. FCA is similar to principal component analysis but using categorical variables (in our case microsatellite genotypes) [57]. To estimate individual ancestries for the SNP data (dataset II), we used ngsAdmix v. 3.2 [58] with K ranging between 2–10 for 100 replicates using default parameters, except for tolerance for convergence (-tol 1 × 10^−6^), log likelihood difference in 50 iterations (-tolLike50 1 × 10^−3^), and a maximum number of EM iterations (-maxiter 10,000). A pairwise genetic distance matrix was computed directly from the genotype likelihoods using ngsDist v.1.0.2 [59] and was then used for Multidimensional Scaling (MDS) using the R package *cmdscale*. 

The proportion of heterozygote sites per individuals was calculated with ANGSD to compute the unfolded global estimate of the Site Frequency Spectrum (SFS) for dataset I. The observed fraction of heterozygous sites was calculated as the ratio between the number of heterozygotes and the total number of sites with information. 

We used NEWHYBRIDS v. 1.1 [60] to estimate the posterior probability of each individual belonging to one of the parental species, F1 hybrids, F2 or backcrosses between F1 and each parental species, based on their allele frequencies. The analysis was run using the default genotype proportions, uniform prior option, burn-in period of 50,000 iteration and 300,000 MCMC sweeps. With the SNP data, we used individuals from FUN and GUA (sympatric populations) for both species (39 individuals: 17 *K*. *ocellatus*, 22 *K*. *hermaphroditus*). We called SNPs with the same parameters described above using ANGSD, but this time with no missing data allowed and selected those (3108 SNPs) with the highest pairwise F_ST_ values between species. Pairs of SNPs with significant LD were removed and randomly replaced with other SNPs to complete a dataset of 200 SNPs (the upper limit of NEWHYBRIDS). We then ran NEWHYBRIDS v.1.1 with the same parameters described for the microsatellites to investigate the posterior probability of each individual to belong to one of the six hybrid classes.

## 3. Results

Both microsatellite genotypes and SNPs confirmed the hybridization between the outcrossing and SI *K. ocellatus* and the predominantly selfing SC *K. hermaphroditus* in two hybrid zones (FUN and GUA in Figure 2) in Southeast Brazil (Figure 3).

Altogether, microsatellites and SNPs identified 12 individuals with admixed genomes between *K*. *ocellatus* and *K*. *hermaphroditus*, seven of those from Fundão in Guanabara Bay (FUN 08, 11, 13, 26, 41, 43, 47) and five from Guaratiba in Sepetiba Bay (GUA 09, 17, 20, 24, 62) all of them sampled in 2017 (Figure 3a,b). The only disagreement between microsatellite and SNP data occurred with one individual from Fundão (FUN 26), that was classified as a pure *K. hermaphroditus* individual with the microsatellites but showed evidence of genetic admixture with SNPs data (see results below). Five individuals identified as hybrids by microsatellite data were not included in the SNPs analyses as they failed to produce our threshold of reads for the GBS library (cut-off ≥ 500 k reads) two from FUN (FUN 13, 41) and three from GUA (GUA 20, 24, 62) (Appendix A). The individual ancestry analyses (STRUCTURE for microsatellites and ngsAdmix for SNP data) supported the identification of the hybrids (Figure 3a,b). At K = 3 (the most likely K according to Evanno’s ΔK method) indicated that all *K*. *ocellatus* were assigned with nearly 100% probability to one cluster, while *K. hermaphroditus* individuals were assigned with nearly 100% probability to another cluster, apart from the subset of the divergent FUN and GUA fish, with admixed genetic backgrounds of both species. The third cluster consisted of the southernmost *K*. *ocellatus* individuals (from the allopatric populations SFR and FLO), reflecting the deep genetic structuring previously found for this species, with isolated populations occupying discontinuous patches of mangrove forests between Southeast and South in Brazil [33]. For both microsatellite and SNP data, the other K values tested indicated admixture between *K. ocellatus* and *K. hermaphroditus* genomes in the subset of FUN and GUA individuals (Appendix A).

Overall, 12 out 145 individuals analyzed (8.2%) had admixed ancestry of *K. ocellatus* and *K. hermaphroditus*. However, when considering only the sympatric populations FUN and GUA, the proportion of admixed individuals increases to 12.5%. Finally, when considering only contemporary sympatric populations from the most recent sampling effort (FUN and GUA from 2017, as no admixed individuals have been detected in the GUA microsatellite dataset from 2007) the proportion of admixed individuals in FUN and GUA in 2017 increases to 19.3%. Of these, eleven of the 30 individuals with *K. ocellatus* mtDNA (36.6%) had admixed nuclear genomes, while only one (3.1%) of the 32 with *K. hermaphroditus* mtDNA had evidence of admixture with *K. ocellatus* at the nuclear genome (Table 1; Figure 3a,b).

The WISO hypothesis predicts that hybridization should be most likely between males of the SI species *K. ocellatus* and hermaphrodites of the SC species *K*. *hermaphroditus*. Accordingly, F1 hybrids should predominately have *K. hermaphroditus* mtDNA haplotypes. Contrary to this prediction, NEWHYBRIDS analysis with both microsatellite and SNP data revealed seven F1 hybrids (FUN 08, 11, 13, 47, 48; GUA 09 and 62), all of them with mtDNA of the outcrossing species *K. ocellatus.* This result reveal that the most prominent direction of hybridization was between the outcrossing and SI *K*. *ocellatus* hermaphrodites and the SC *K*. *hermaphroditus* males. NEWHYBRIDS results also revealed four (FUN 41; GUA 17, 20, 24) backcrosses between *K*. *ocellatus* and a F1 hybrid, and one as a backcross between *K*. *hermaphroditus* and a F1 hybrid (FUN 26, only indicated with SNP data). This partial evidence of asymmetrical direction of backcrossing is consistent with the prediction that introgression is most common from the SC into the SI species. No individual, regardless of the genetic marker, was identified as a F2 (Figure 3c,d). All hybrids, except for FUN 26, had a mtDNA *cox1* haplotype typical of the outcrossing SI species *K*. *ocellatus*. In terms of overall genomic variation, both factorial correspondence and multidimensional scaling analyses positioned the F1s and backcrosses in between the parental genotypes along the first axes, while the variation along the second axis separated the *K. ocellatus* populations from south and southeast Brazil (Figure 3e and Appendix A).

## 4. Discussion

Differences in mating systems are expected to influence the extent, direction, and the structure of hybrid zones involving SI and SC species [5,19,25]. Given the strong reproductive isolation of highly selfing species, such as *K*. *hermaphroditus* [30,35], theory predicts that there would be low hybridization rates in hybrid zones involving SI × SC species [5]. Contrary to this prediction, we found relatively high hybridization rates between *K*. *ocellatus* (obligately outcrossing) and *K*. *hermaphroditus* (predominantly selfing), with admixed individuals representing approximately 20% of the individuals in the coexisting populations. To our knowledge, this represents the first case of hybridization between vertebrate species with different mating systems. Contrary to the expectations of the ‘SI × SC rule [11] our data indicates that all seven F1 hybrids were sired by the rare males of predominantly selfing species *K*. *hermaphroditus.* Given the difficulties to detangle mating opportunities and hybrid viability in natural populations, we acknowledge that our results do not represent a formal test of the weak inbreeder/strong outbreeder (WISO) hypothesis [11], which would require an experimental manipulation of mating opportunities, a task that has been proven particularly challenging for mangrove killifishes in laboratory conditions [36,44,61]. Although our sample size of backcrosses was small, those were mostly asymmetrical, showing a bias (four out of five) towards backcrosses between F1s and the SI parental species, *K. ocellatus.* A higher rate of backcrossing from the SC to SI species is consistent with the asymmetrical direction of introgression commonly observed in plant hybrid zones with different mating systems [5,6,7,19].

In most killifish species, egg fertilization occurs externally [62]. However, in the selfing mangrove killifishes, most of the eggs laid externally by the hermaphrodites are already fertilized internally via selfing [36,37]. This reproductive trait, similar to the ‘prior selfing’ in plants [22], leaves a limited window of opportunity for fertilization via outcrossing by either the (rare) conspecific or heterospecific males. Given that evidence suggest that outcrossing in the predominantly selfing mangrove killifishes only happens between male and hermaphrodites [34,44], outcrossing opportunities involving this species depend both on the density and mating activity of sexually active males as well as the availability of unfertilized eggs [36]. Despite extensive historical sampling, particularly in southeast Brazil, males of the SC *K. hermaphroditus* seem to be rare or even absent in some populations [35,63]. In fact, males of this species were only discovered recently [31,32]. The WISO hypothesis predicts that SI gametes should represent a strong barrier to fertilization for gametes of SC species. This prediction was not supported by our results, as all F1s were sired by males of the selfing *K. hermaphroditus*. With our data, we could not formally distinguish whether this pattern was caused by the potential differential reproductive barriers caused by gamete incompatibilities (WISO) or due to differential mating opportunities in the wild for crosses of opposite directions to happen. However, interspecific laboratory crosses of *Caenorhabditis* nematode species also revealed the opposite pattern to the SI × SC rule [21], suggesting that there are exceptions to this rule, as previously observed in plant hybrids [9,15]. In plants, it has been suggested that exceptions to the SI × SC rule occur in species where SC recently evolved from SI (Brandvain and Haig; 2005). Yet, the timing of the transition from SI to SC within *Kryptolebias* genus is unknown. Future research is also needed to investigate whether occasional outcrossing among divergent selfing lineages of *K. hermaphroditus* [35] may maintain competitiveness of the SC gametes (and therefore higher levels of sperm-egg conflict), possibly attenuating the effects of relaxed selection on sperm performance and other sexually selected traits due to prolonged selfing [a common effect of the selfing syndrome, 28]. 

Hybrids between highly divergent species (such as *K*. *ocellatus* and *K*. *hermaphroditus* [33,43]) are expected to harbor many genetic incompatibilities caused by the gradual accumulation of divergent alleles [64,65]. Although higher-resolution genomic data is still needed to investigate the possibility of ancestral introgression in this system, the absence of admixture in the one of the hybrids zones in recent collections (GUA in 2007) and the lack of further hybrid generations beyond F1s suggest that the hybrid zones found here are relatively recent, and the reproductive isolation between *K. ocellatus* and *K. hermaphroditus* is strong, despite being incomplete. The mangrove forests (FUN and GUA) where hybridization was found are situated within Guanabara and Sepetiba Bays, respectively. Those bays surround the Rio de Janeiro municipality and are amongst the most polluted Brazilian estuaries [66]. Water pollutants are known to disrupt physiology, mate choice and reproductive isolation between coexisting species [67,68]. The fact that we have not found evidence for hybridization in GUA samples from 2007, but found it in high rates in 2017, suggests that a recent environmental change (possibly human-induced) may have disrupted some of the reproductive barriers between syntopic *K*. *hermaphroditus* and *K. ocellatus* populations. Further sampling in the area is urgently needed to shed light the potential links between hybridization rates and environmental stressors across a larger number of sympatric populations.

The contrasting nature of genetic load between selfers and outcrossers [25] could provide a potential explanation for the observed structure and direction of backcrossing found in our study. While outcrossers are expected to have many highly deleterious recessive alleles, selfers are expected to purge those and have many mildly deleterious codominant alleles [5]. Selfed progeny of F1s between selfing and outcrossing species are expected to have low fitness given the increased exposure of an outcrosser’s recessive load in homozygosity. Although we cannot fully rule out the possibility that F1s between *K. ocellatus* and *K. hermaphroditus* are able to self-fertilize, our results indicate that they are at least able to outcross via backcrossing. Therefore, it might be possible that high genetic load (with many mildly deleterious codominant alleles) originating from selfers via introgression is quickly removed by natural selection from the outcrossing population. This would result in hybrid zones only containing early generation hybrids [5], with some individuals of *K. ocellatus* still bearing self-derived ancestry given recent backcrossing, but with no evidence of ancestral introgression. Further research is needed to investigate whether *K. ocellatus* individuals with self-derived ancestry show any reduction of fitness compared to ‘pure’ *K. ocellatus* individuals.

Another property of the hybrid zone between *K. ocellatus* and *K. hermaphroditus* is its asymmetry (unidirectional hybridization, and partial evidence for asymmetrical backcrossing). Asymmetrical hybridization suggests differential strengths of barriers to gene flow according to the direction of hybridization [5]. The influence of mating system differences vs selection and/or other reproductive barriers on hybridization zones are hard to disentangle in natural populations [6,69]. Although we cannot rule out the possibility of postzygotic unilateral incompatibility [also known as the ‘Darwin’s corollary to Haldane’ rule, 26], the asymmetries in the direction of hybridization and the small but biased ratio of backcrossing from SC to SI suggest that differential mating, or more precisely, differential fertilization opportunities (the ‘mating opportunity’ hypothesis) may have also had a role in the structure of the hybrid zones. This hypothesis is substantiated by some life-history details of the species involved: (i) the rare males of the SC *K. hermaphroditus* species are more likely to find heterospecific than conspecific unfertilized eggs available for fertilization, given the ‘prior selfing’ nature of its conspecific hermaphrodites; (ii) backcrossing, although small, was bidirectional, revealing that at least some of the offspring between F1s and *K. hermaphroditus* are viable. Potential male F1s are more likely to find fertilizable eggs laid by the SI *K. ocellatus* than the SC *K. hermaphroditus*, while potential female F1 individuals would also be more likely to be fertilized by the more abundant males of the SI *K. ocellatus* than the rare males in SC *K. hermaphroditus.* Although we observe a slight bias of backcrosses from SC to SI (4:1 ratio), the overall small number of backcrosses calls for further research on the role of mating opportunities on backcrossing asymmetry. In addition to the increased fertilization opportunity, the propensity of males of the SC *K. hermaphroditus* to fertilize heterospecific eggs could be intensified by its patterns of disassortative mating. In the SC *K. marmoratus,* sister species of *K. hermaphroditus* [30,43], males tend to associate with genetically dissimilar hermaphrodites [70]. If this pattern of disassortative association holds true for *K. hermaphroditus* in a heterospecific context, both egg availability and behavioral preferences may have influenced the asymmetric direction of hybridization observed here. Further studies are needed to elucidate the patterns of assortative mating among *K. ocellatus*, *K. hermaphroditus,* and their hybrids.

## 5. Conclusions

Investigating the influence of different prezygotic barriers in hybrid zones is crucial for the understanding of the evolutionary consequences of hybridization, the evolution of reproductive isolation, and ultimately, speciation. Although the role of contrasting mating systems in hybrid zones has been largely explored in plants [5,7,11,20], empirical studies in animals are scarce. Our study contributed to this understanding by exploring, for the first time in a vertebrate system, the effect of different mating systems (predominantly self-fertilizing vs outcrossing) in natural hybrid zones. We observed hybridization between a predominantly self-fertilizing and an outcrossing fish species that differed in several aspects from what has been observed in other hybrid zones between SI and SC taxa, particularly in plant systems. First, hybridization rates were relatively high for a highly selfing taxa with prior selfing such as *K. hermaphroditus.* All F1 hybrids were sired by the selfing species, providing a potential exception for the SC × SI rule in a natural animal system. Although these findings partially challenge the predictions of the WISO hypothesis, further studies are necessary to detangle the role of mating opportunities and/or hybrid viability according to the crosses directions.

While hard to disentangle from other reproductive barriers, the asymmetry in hybridization and backcrossing found here suggest that fertilization opportunities may have had a strong role in shaping the direction of gene flow between the mangrove killifish species. In addition to influencing the degree of reproductive isolation between sympatric species, asymmetrical introgression can also affect the reproductive compatibility within species between allopatric and sympatric populations, if the latter are experiencing asymmetrical introgression [71]. Further studies are needed to investigate whether the asymmetric gene flow of *K. hermaphroditus* genome into *K. ocellatus* in the hybrid zones has impacted the reproductive isolation between allopatric vs sympatric *K. ocellatus* populations.

## Figures and Tables

**Figure 1 genes-12-01486-f001:**
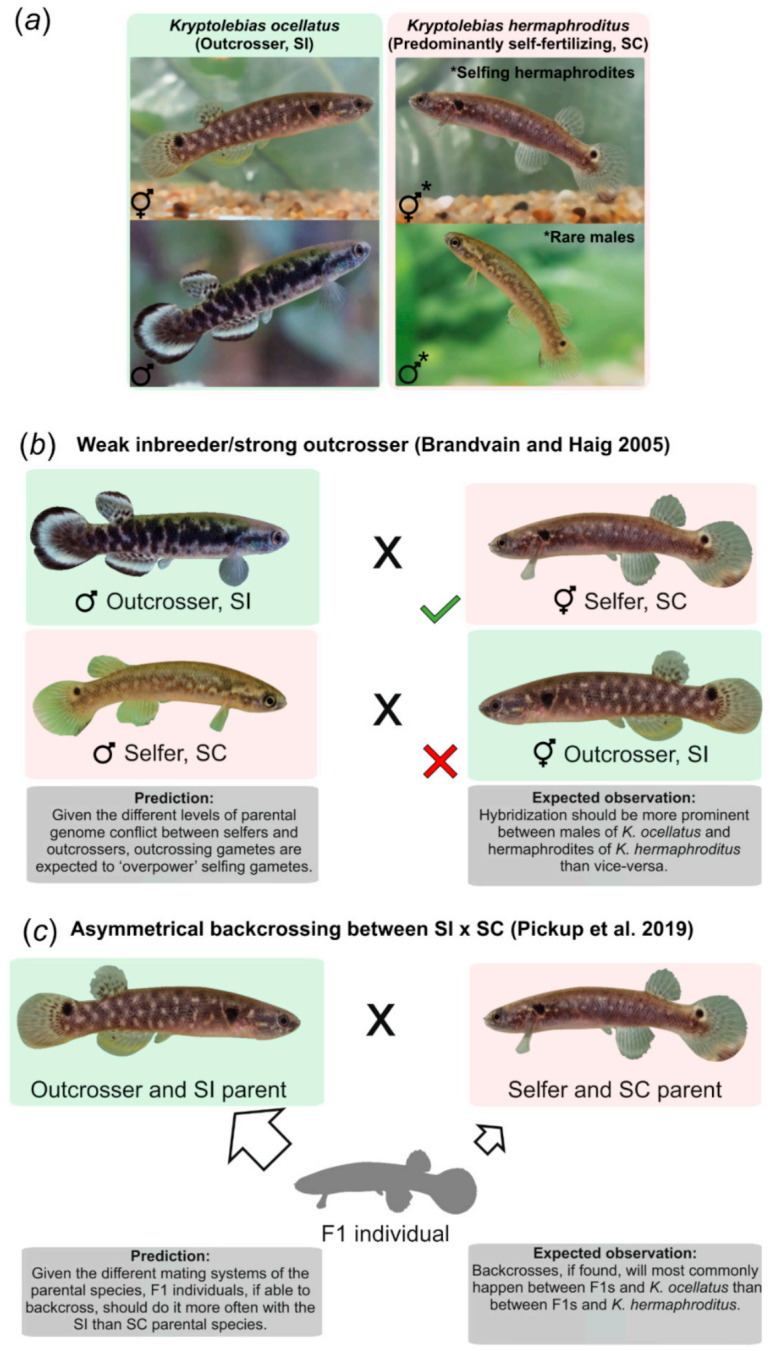
Mangrove killifish species and schematic representation of predictions and expected observations. (**a**) Mangrove killifish species with respective mating systems. Mating system classifications for *Kryptolebias ocellatus* and *K. hermaphroditus* mating were retrieved from [33,35], respectively. ‘SI’ refers to self-incompatible. ‘SC’ denotes self-compatible. (**b**) Representation for the weak inbreeder, strong outcrosser hypothesis (WISO) [11]. This hypothesis predicts that outcrossers have higher potential for genome conflict given the higher likelihood of crossing between divergent genomes when compared to selfers. This higher potential for genome conflict should reflect on the level of gametes competitiveness with outcrossers gametes being able ‘overpower’ selfers gametes [11]. Thus, the prediction is that hybridization between species with different mating systems should occur more often between males of the outcrossing species with hermaphrodites of selfers (as competitive sperm should overcome the potential reproductive barrier imposed by the less competitive selfers eggs). The opposite direction is less likely to happen as outcrossers’ eggs should impose a stronger barrier for the less competitive sperm from selfers. (**c**) Representation for asymmetrical backcrossing between SI and SC hypothesis [5]. Given the differences between parental mating systems, if F1 individuals are viable and able to backcross, the direction of backcrossing should be biased towards the SI parent (higher gametes transmission) rather than the SC (gametes partially fertilized through selfing) parent.

**Figure 2 genes-12-01486-f002:**
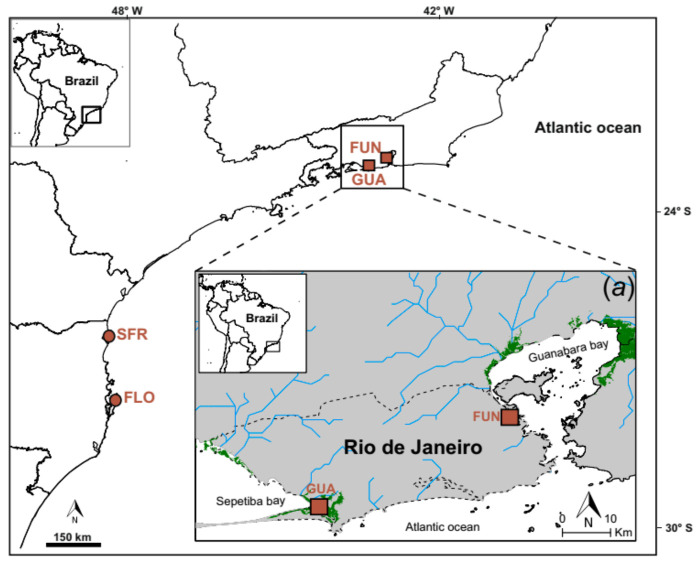
Sampling sites for the individuals included in the genetic analysis. Site names and details included in Table 1. Brown squares represent syntopic populations of *Kryptolebias ocellatus* and *K*. *hermaphroditus* and circles represent populations where only *K*. *ocellatus* was found. (**a**) Detailed map of the Rio de Janeiro municipality and its surrounding bays and mangroves.

**Figure 3 genes-12-01486-f003:**
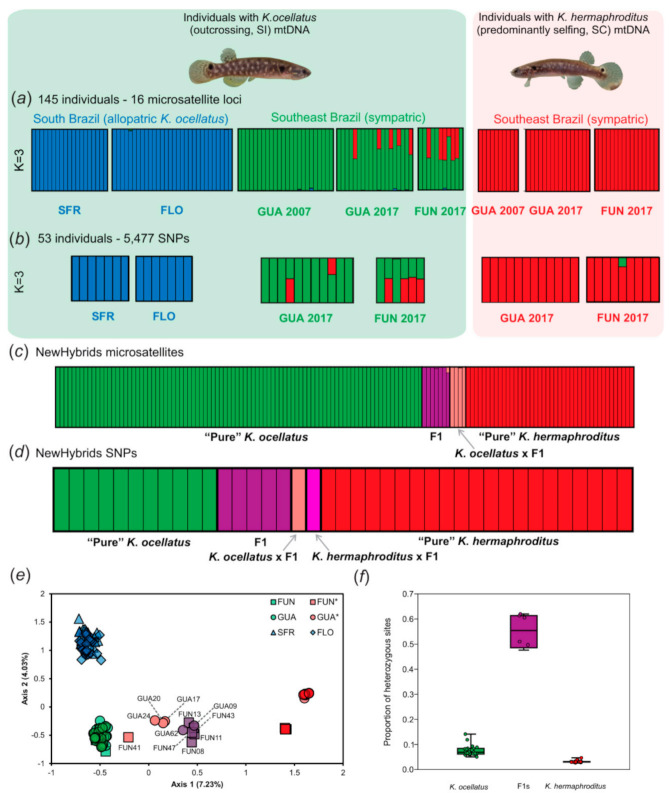
Genetic analyses for the hybrid zone between *Kryptolebias ocellatus* and *K*. *hermaphroditus* in Southeast Brazil. Admixture plots showing the genetic clusters (K = 3) for the (**a**) 16 microsatellites loci amplified in *K. ocellatus* and *K*. *hermaphroditus* according to STRUCTURE and (**b**) for the 5477 SNPs according to ngsAdmix (see Methods for details). Each individual is represented by a bar, and each color represents a genetic cluster. (**c**,**d**) NEWHYBRIDS individual classification into hybrid classes (*K*. *ocellatus*, *K*. *hermaphroditus*, F1, F2, *K*. *ocellatus* × F1 and *K*. *hermaphroditus* × F1) using microsatellite and SNPs data. (**e**) Factorial correspondence analysis using microsatellites data for all *K*. *ocellatus* and *K*. *hermaphroditus* individuals colored (green and blue for *K. ocellatus* from Southeast and South Brazil, respectively; red for *K. hermaphroditus*) and shaped according to sampling sites (squares, circles, triangles, and diamonds represent individuals from FUN, GUA, SFR and FLO, respectively). Hybrid individuals (see Results) are highlighted with their respective labels and colored (purple for F1s, pink for backcrosses) according to the hybrid class indicated by NEWHYBRIDS analysis. Sampling locations with asterisks represent *K*. *hermaphroditus* individuals. (**f**) Proportion of heterozygous sites between *K*. *ocellatus*, *K*. *hermaphroditus* and F1 hybrid individuals.

**Table 1 genes-12-01486-t001:** Sampling localities and sampling sizes for *Kryptolebias ocellatus* (Koce) and *Kryptolebias hermaphroditus* (Kher) in south and southeast Brazil. Microsatellites information for Guaratiba (GUA in Figure 1) was extracted from individuals sampled in two different sampling periods, 2017 and 2007. ‘Msats’ refers to sampling sizes for microsatellites. ‘SNPs’ refers to final sampling sizes using single-nucleotide polymorphisms. ‘Reference for Msats’ refer to references from which microsatellite data was extracted. RJ, Rio de Janeiro State; SC, Santa Catarina State. Asterisks denote sympatric populations.

Sample ID	Location	Latitude	Longitude	Msats (Koce/Kher)	SNPs (Koce/Kher)	Reference for Msats
FUN 2017*	Fundão mangrove, Rio de Janeiro, RJ	22°52′2.50″ S	43°13′27.50″ W	11/16	6/10	Berbel-Filho et al. [33]/This study
GUA 2017*	Piracão mangrove, Guaratiba, RJ	23°0′1.90″ S	43°34′51.50″ W	19/16	11/12	Berbel-Filho et al. [33]/This study
GUA 2007*	Piracão mangrove, Guaratiba, RJ	23°0′1.90″ S	43°34′51.50″ W	24/10	-/-	Tatarenkov et al. [40]
SFR	Linguado channel, São Francisco do Sul, SC	26°22′0.02″ S	48°39′58.40″ W	19/-	7/-	Berbel-Filho et al. [33]
FLO	Rio Ratones estuary, Florianópolis, SC	27°28′3.84″ S	48°29′33.76″ W	30/-	7/-	Berbel-Filho et al. [33]
Total				103/42	31/22	

## Data Availability

FastaQC files for GBS library can be accessed at NCBI (accession PRJNA563625). Sequence and microsatellite data are available in Appendix A. All scripts used in the project are available at: https://github.com/waldirmbf/BerbelFilho_etal_KryptolebiasHybridisation.

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
