# Peer review of "Against the Odds: Hybrid Zones between Mangrove Killifish Species with Different Mating Systems"

_genes, 2021, doi:10.3390/genes12101486_

Round 1

Reviewer 1 Report

This was an interesting study on hybridization between outcrossing and selfing mangrove killifish species that found hybridization directions contrary to predictions (involving outcrossing maternal parents) and introgression. Below, I give some minor suggestions for improvement and clarity.

1) In this study, both microsatellite and SNP data were collected analyzed. I would like to see the authors provide some justification in the introduction as to why both types of data are needed and how they provide complementary, but different, information. Why not just use the SNP data?

2) In lines 211-213, the authors describe two genomic data sets one with ~500,000 and one with ~5,000 SNPs. I didn't understand the differences between these data sets or why it seems only the second one was presented in the results.

3) In lines 99-101, the authors state that K. ocellatus is a primarily outcrossing species despite the presence of hermaphrodites. I think the previous evidence for this could be presented in more detail as it is very important to the context of the paper. Also, in these killifish species do hermaphrodites ever fertilize other eggs from another individual or do they lack the structures for external fertilization?

4) Can the number of alleles  per population and other standard information for the microsatellites be presented as a supplementary table?

5) In line 62, a mention of "SI x SC" rule test in nematodes is made. As this appears to be the only other animal example, could be described in more detail?

6) Line 75 makes a reference to "UI" what is this abbreviation. Did the authors mean "SI"?

7) In line 329, the authors refer to "relatively high" hybridization rates. Relative to what? Perhaps the authors can compare to hybridization rates in other fish species?

8) In line 369, the two species are described as "highly divergent." I did not see any genetic divergence or genetic distance presented. What was the mean FST between the species?

9) The discussion of genetic load was a little confusing. For example, in line 389, the authors describe "high genetic load originating from selfers" but it seemed that the previous few sentences had suggested selfers can purge their genetic load.

10) In lines 178, the GBS data is described as "methylation sensitive." I am unfamiliar with this approach. Why was is used?

Author Response

All my responses are in the attached file.

Reviewer 2 Report

In this manuscript the authors sample wild sympatric populations of two different mangrove killifish. These fish differ in their mating system; one is a predominant self-fertilizer and the other is an outcrosser. The authors use microsatellite data and whole genome resequencing to resolve the frequency of hybridization between the two as well as the general direction of gene flow. F1s all came from a single cross direction which the authors interpret as both evidence against WISO and evidence for differences in mating opportunities. While the data and analyses are solid I found the conclusions to be overreaching and the paper difficult to read.

My main issue is that the authors are conflating mating opportunity and offspring viability in a lot of the text. With their experimental design they cannot actually disentangle the two. The viability prediction (based on WISO) is that most hybrids should be from crosses between selfing hermaphrodites and outcrossing males (Figure 1). However, the mating opportunity predicts that crosses between the outcrossing females and selfing hermaphrodites should be most common. The authors state that in the hermaphroditic species most eggs are already fertilized before they are laid (Lines 94-98) so most mating opportunities should be in the opposite direction. Given that no laboratory crosses were performed and the proportion of unfertilized eggs laid by the selfing species is unknown, they cannot actually truly test the WISO hypothesis with this data. For example, the viability of selfing hermaphrodites and outcrossing males crosses could be only 10% but the mating opportunity in the other cross direction could be 0%. This would give similar results (given that enough hybridization occurs).

That said I think that the paper is laid out well but the authors should be crystal clear that they cannot actually test the WISO hypothesis with this design. The intro and discussion should be reworked to clarify this and more concretely separate mating opportunity from cross viability. While the text does discuss both, it flips back and forth a lot, which is confusing for the reader and makes it unclear what the authors are actually testing. Furthermore, it seems like the F1 hypothesis is based on WISO and the backcross hypothesis is based on mating opportunity? I assume the WISO hypothesis also makes predictions about backcrosses? It seems weird to shift between the two. Additionally, stating that backcrossing is asymmetric with a 4:1 ratio and 5 samples is not supported and the discussion of this should be removed.

Other comments:

-A table with full sample information (year collected, site, type of sequencing, hybrid status) should be given in the supplementary.

-The lack of hybrids in GUA in 2007 should be discussed. The fact that hybrids occurred within 10 years at such a high rate is likely an artifact of some kind.

-A lot of the discussion is overly speculative and should be toned down (369-392).

Minor comments:

-The figure 3e legend should include information about what the shapes and colors represent (i.e. which color and which shape corresponds to what kind of individual).

-There is no information about the number of libraries sequenced per lane.

-Supplemental Table S1 should distinguish between putative F1s and backcrosses

-Does 3F only include F1s? If there are putative backcrosses in there, they should be graphed separately.
